# Indigenous Medicinal Plants Used in the Management of Diabetes in Africa: 5 Years (2019–2024) in Perspective

**DOI:** 10.3390/plants13141898

**Published:** 2024-07-10

**Authors:** Ebenezer Kwabena Frimpong, Nokukhanya Thembane, Sphamandla Hlatshwayo, Mlungisi Ngcobo, Nceba Gqaleni

**Affiliations:** 1Traditional Medicine Laboratory, School of Nursing and Public Health, College of Health Sciences, University of KwaZulu-Natal, Durban 4041, South Africa; 210505551@stu.ukzn.ac.za (N.T.); thembane@mut.ac.za (S.H.); ngcobom3@ukzn.ac.za (M.N.); gqalenin@ukzn.ac.za (N.G.); 2Department of Biomedical Sciences, Faculty of Natural Science, Mangosuthu University of Technology, Durban 4026, South Africa; 3Africa Health Research Institute, 3rd Floor K-RITH Tower Building, Nelson R. Mandela School of Medicine, Durban 4001, South Africa

**Keywords:** Amaryllidaceae, Fabaceae, indigenous medicinal plants, diabetes mellitus

## Abstract

(1) Background: The utilization of medicinal plants in the management of diabetes is of great importance to the health of the indigenous population on the African continent. The high cost of orthodox conventional medicines coupled with the perceived side effects encourages the high patronage of indigenous medicinal plants in the management of this metabolic disorder. We conducted a review on the indigenous African medicinal plants that could be useful in preclinical and clinical trials in the field of diabetes mellitus research. (2) Methods: Data were mined from research articles published and associated with the use of medicinal plants in the management of diabetes on the African continent (from January 2019 to March 2024). Literature from ethnobotanical studies on the African continent was searched from the Google Scholar, SCOPUS, Medline, Web of Science and PubMed databases. We employed the following keywords: “indigenous plants”, “diabetes in Africa” and “antidiabetic effect of medicinal plants”. (3) Results: A total of 52 medicinal plants belonging to 31 families were recorded in this study. Amaryllidaceae (14.28%), Fabaceae (9.52%) and Asteraceae (9.52%) were the most cited plant families. The frequently used indigenous medicinal plants on the African continent were *Allium sativum* L. (*n* = 6), *Olea europaea* L. (*n* = 6), *Azadirachta indica* A. Juss (*n* = 5), *Allium cepa* L. (*n* = 5) and *Moringa oleifera* Lam. (*n* = 5). The highly cited parts of the plant used in the management of diabetes were the leaves (45.55%), bark (14.44%) and bulbs (12.22%). The preferred methods of the preparation of herbal medicines were decoction (45.23%) and infusion (25%). Oral (52.32%) was the preferred route of the administration of herbal medicine on the African continent. (4) Conclusions: The data revealed some similarities in the way countries on the African continent manage diabetes. This interesting observation will assist in our quest towards obtaining a standardized protocol using indigenous medicinal plants to combat diseases on the African continent.

## 1. Introduction

Diabetes mellitus (DM) is one of the major diseases associated with morbidity and mortality on the African continent. The literature suggests that about 19 million adult people in Africa have diabetes [1]. More importantly, a study conducted by Shaw et al. [2] revealed that the Asia and Africa regions present the greatest risks because the prevalence rate could rise threefold in the not-too-distant future. DM is categorized as a triangular disorder that involves complex interactions among fats, carbohydrates and pancreatic enzymes such as glucagon and insulin. The energy that the cells in humans utilizes to function optimally originates from proteins, carbohydrates and lipids. The glucose released into circulation from the liver is the product of carbohydrate ingestion. Pancreatic *β* cells release the hormone insulin so that it can interact with specific cellular receptors to ensure the entry of glucose into the cells. The insulin obtained from the pancreatic *β* cells facilitates the entry of glucose into the cells for its energy conversion [3]. According to the World Health Organization (WHO), DM is characterized by a fasting plasma glucose value of about 126 mg/Dl (7 mmol/L), or a higher value recorded on two separate tests [4]. The four major classifications of diabetes include Type 1, Type 2, gestational and diabetes specifically induced by drugs, cystic fibrosis and latent-action disease in adults [5]. The major symptoms of diabetes include blurred vision and polyuria [6], and the major complications of the disease are atherosclerosis, chronic kidney disease, nerve damage, retinopathy and heart attack [7]. Diabetes is managed with insulin, pharmacotherapy and diet [8,9]. It is significant to point out that even though orthodox conventional medicine (OCM) assists in the management of diabetes, its use is associated with side effects such as heart and liver diseases and skin rashes [10]. The side effects associated with the use of OCM (Table 1 and Table 2) to manage diabetes encourage patients to rely on African traditional medicine (ATM) [11]. The high patronage of ATMs, mainly herbal mixtures, in the management of diabetes on the African continent is due to their affordability and their perceived lack of side effects [12].

A study revealed that medicinal plants play an important role in the management of diabetes [13]. These medicinal plants are effective in the management of diabetes because of their phytochemical constituents, such as flavonoids, glycosides and carotenoids, which act as anti-hyperglycemic agents [14]. The literature suggests that the antidiabetic potential of the phytochemical constituents present in medicinal plants could be due to their effect on the glycogenesis stimulation coupled with the decrease in glucose absorption and the activation of *β* cells to release insulin to assist in glucose use in patients [15,16]. Furthermore, these phytochemical constituents associated with medicinal plants could assist in the termination of oxidative stress on *β* cells. These phytochemical constituents can also restore impaired *β* cells [17,18].

Considering the use of medicinal plants in the management of diabetes in Africa, it is important to assist the scientific community with the most common plants for further studies in the quest to mitigate the effects of the disease in our societies. This review aims to provide information about the medicinal plants used in the management of diabetes on the African continent over the last 5 years (2019–2024).

**Table 1 plants-13-01898-t001:** Orthodox conventional medicines used in the management of DM (oral hypoglycemic drugs).

Classification	Drugs and Their Synonyms	Mechanism of Action	Pharmacological Effects	Side Effects
Sulfonylurea derivatives	Glibenclamide, glicaside	Assist in the stimulation of the endogenic insulin production of β cells of the pancreas [19]	Hypoglycemic, hypolipidemic	Weight gain [20]
Biguanide derivatives	Metformin, buformin	Overpowering of gluconeogenesis in the liver, lipolysis stimulation, uptake of glucose utilization by peripheral tissues and reduction in glucose absorption in the gastrointestinal tract [21]	Hypoglycemic, fibrinolytic and anorexigenic	Gastrointestinal disturbance, lactic acidosis [22]
Meglitinides	Repaglinide, nateglinide	Helps in the reduction in the blood glucose levels by stimulating the production of insulin [23]	Hypoglycemic	Dizziness and headache [24]
Thiazolidinediones	Pioglitazone, rosiglitazone	Assist in the activation of the nuclear peroxisome proliferator receptor-gamma (PPAR-gamma), thereby increasing the insulin action and decreasing the blood glucose [25]	Anti-hyperglycemic, hypoglycemic	Weight gain, cardiac hypertrophy, pulmonary edema [26,27,28]
Alpha glucose inhibitors	Acarbose, miglitol	Assist in the reduction in the blood glucose levels via the delay of digestion and the absorption of multifaceted carbohydrates [29]	Hypoglycemic	Gastrointestinal disturbance [30]

**Table 2 plants-13-01898-t002:** Orthodox conventional medicines applied in the management of DM (Insulin).

Classification	Drugs and Their Synonyms	Mechanism of Action	Pharmacological Effects	Side Effects
**Human origin**				
Short-term acting	Humalog, Actrapid	Assist in the regulation of carbohydrate metabolism, the promotion of the transportation of glucose inside the cells and its assimilation by the tissues; helps in the increment of glycogenesis, protein and fatty acid synthesis [31]	Hypoglycemic effect	Mild burning sensation in the nostrils of patients [32]
Medium-term acting	Humulin, Monotard	Assist in the regulation of carbohydrate metabolism, the promotion of the transportation of glucose inside the cells and its assimilation by the tissues [33]	Hypoglycemic effect	Vomiting, dizziness [34]
Long-term acting	Ultratard	Assists in the regulation of the carbohydrate metabolism, the promotion of the transportation of glucose inside the cells and its assimilation by the tissues; helps in the increment of glycogenesis, protein and fatty acid synthesis [35]	Hypoglycemic effect	Incidence of myocardial infarctions [36]
**Animal origin**				
Short-term acting	Monoinsulin	Assists in the regulation of the carbohydrate metabolism, the promotion of the transportation of glucose inside the cells and its assimilation by the tissues; helps in the increment of glycogenesis, protein and fatty acid synthesis [37]	Hypoglycemic effect	Severe hypoglycemic episodes [38]
Medium-term acting	Insulin Semilente, B-insulin-S	Assist in the regulation of the carbohydrate metabolism, the promotion of the transportation of glucose inside the cells and its assimilation by the tissues; helps in the increment of glycogenesis, protein and fatty acid synthesis [39]	Hypoglycemic effect	Lipoatrophy on the patient’s thighs [40]
Long-term acting	Ultralente	Assists in the regulation of the carbohydrate metabolism, promotion of the transportation of glucose inside the cells and its assimilation by the tissues; helps in the increment of glycogenesis, protein and fatty acid synthesis [37]	Hypoglycemic effect	Increase in the body weights of patients [41]

## 2. Methods

### 2.1. Selection of Publications Employed in This Study

We carried out the research by employing a combination of the following words: “indigenous medicinal plants”, “diabetes in Africa”, “antidiabetic effect of medicinal plants”, “hypoglycemic activity of medicinal plants” and “indigenous preparations of herbal mixtures”. Employing the Google Scholar, SCOPUS, Medline, Web of Science and PubMed databases, published articles were accessed for possible inclusion in this study. 

### 2.2. An Overview of the Selection Procedure Used in This Study

Generally, the inclusion criteria were published articles written in the English language (from January 2014 to March 2024), the provision of the scientific name of the medicinal plant and the location of the study area on the African continent. The exclusion criteria involved articles not written in the English language, literature or systematic review articles, work related to the use of OCM in the management of diabetes, abstract-only researched articles and case reports, books, manuals and related studies reporting animal-/mineral-based African traditional medicine (ATM) employed in the management of diabetes. The authors did employ a four-step selection criterion process to identify potential articles relevant to this study. To begin with (Step 1), the relevance of the identified published articles was assessed based on their captions and recorded on a data sheet. Secondly, abstracts of the published studies were assessed for their conformity to the inclusion criteria. Thirdly, the authors obtained full-length texts from the identified published articles based on the knowledge acquired in Step 2 to make an informed decision as to whether to add or reject these publications in this review. Lastly, the published research articles that satisfied the inclusion criteria mapped out by the authors were collated for this study. A total of 39 research articles were identified and included in this study (Figure 1). The data were scrutinized to extract relevant information, such as botanical names, local names, countries, routes of administration and modes of preparation (Table 3). Employing recognized databases such as the Plant List and Plant ZA databases (including the Global Biodiversity Information Facility database), the botanical names of the medicinal plants captured in this study were validated [42,43]. 

## 3. Results and Discussion

### 3.1. Patterns and Distributions of Medicinal Plants Utilized in the Management of Diabetes in Africa

In this study, we recorded 39 eligible studies (Figure 1) based on our search criteria (Section 2.1) from different regional blocs on the African continent. Our data revealed that the Western African Region (WAR) recorded the highest number of studies (*n* = 13), followed by the Northern African Region (NAR) (*n* = 12), Southern African Region (SAR) (*n* = 7), Eastern African Region (EAR) (*n* = 6) and Central African Region (CAR) (*n* = 1). Regarding the countries that recorded the highest numbers of studies, Nigeria recorded one of the highest (*n* = 8). The other African countries that made the list are as follows: Morocco (*n* = 8), Algeria (*n* = 3) and South Africa (*n* = 4). A total of 52 medicinal plants representing 31 families were included in this review. This study has shown that, notwithstanding the geographical location on the African continent, the use of medicinal plants in the management of diabetes is of great prevalence. Similarly, the application of medicinal plants in the management of diabetes on different continents has been established in other studies [82,83].

### 3.2. Assessed Medicinal Plants and Families Used in Diabetes Management on the African Continent

Our study revealed that the most dominant plant families were *Amaryllidaceae* (14.28%), *Fabaceae* (9.52%), *Asteraceae* (9.52%) and *Meliaceae* (7.14%) (Figure 2). This is largely in agreement with a study conducted by van Wyk (2020), which revealed that the most dominant plant families in sub-Saharan Africa are Fabaceae (567 spp., 156 genera), Lamiaceae (142 spp., 37 genera) and Asteraceae (314 spp., 112 genera) [84]. Interestingly, similar studies around the globe have mentioned the association of these plant families with the management of diabetes in their respective geographical areas [85,86]. Scientific data revealed the antidiabetic potency of the Amaryllidaceae family due to the presence of their secondary metabolites (flavonoids and alkaloids), which assist in decreasing blood glucose levels [87]. The Fabaceae family’s antidiabetic potential is based on their alpha-amylase inhibitory effects in organisms [88]. The Meliaceae family’s antidiabetic potential could also be due to their ability to inhibit alpha-amylase [89]. Bessada et al. [90] reported the antidiabetic potential of the Asteraceae family due to its stronger antioxidant properties, which assist in diabetes management. The most cited plant species in this study were *Azadirachta indica* A. Juss (*n* = 6), *Allium sativum* L. (*n* = 6), *Olea europaea* L. (*n* = 6), *Allium cepa* L. (*n* = 5) and *Moringa Oleifera* Lam. (*n* = 5) (Table 3). It is significant to point out that experimental data confirmed the antidiabetic potential of the most cited plant species in this study. For example, meliacinolin (novel tetranortriterpenoid) extracted from *Azadirachta indica* A. Juss repressed the alpha-amylase and alpha-glucosidase enzyme inhibitory activities in mice with streptozotocin (STZ)-induced diabetes [91]. Similarly, the oral administration of garlic extracts to rats with STZ-induced diabetes improved the serum glucose and cholesterol levels in the experimental animals [92]. The above-explained experimental models using some of the most cited plant species recorded in this study justify the utilization of these indigenous medicinal plants within traditional African communities. However, the issue of the safety and use of medicinal plants should be taken into consideration, as recommended by the WHO policy on alternative medicine [93]. Kharchoufa et al. [94] argued that the perception regarding medicinal plant use in the management of diseases might not be entirely true. This is because of the adverse effects associated with the use of medicinal plants, such as carcinogenicity, teratogenicity and even death [94]. A study revealed that the use of medicinal plants may be toxic when taken together with drugs [95]. Ponnusankar et al. [95] reiterated the need to encourage research-related activities associated with herb–drug interactions. Furthermore, the authors recommended the evaluation of herbal medicines to determine their efficacy before their application in the management of diseases [95].

### 3.3. Commonalities in the Utilization of Similar Medicinal Plants in the Management of Diabetes on the African Continent

This study has shown similarities in the use of certain medicinal plants in the management of diabetes on the African continent (Table 3). For example, *Allium cepa* L., used in the management of diabetes in Ghana [49], is also used in the management of diabetes in Algeria [50] and Uganda [79]. Interestingly, the traditional use of *Allium cepa* L. in the management of diabetes has been well documented in studies conducted in Jordan [96] and India [97]. *Moringa oleifera* Lam., which is utilized in the management of diabetes in Nigeria [54], is also used in Angola [60] and Zambia [53] for similar purposes. Fascinatingly, ethnobotanical studies in Barbados [98] and the Philippines [99] revealed the use of *Moringa oleifera* Lam. by participants in the management of diabetes. This interesting observation (similarities associated with the use of medicinal plants [49,50,53,54,60,79]) is important to assist in the development of standardized health protocols using indigenous medicinal plants for the treatment and management of various diseases on the African continent. The development of a standardized health protocol employing indigenous medicinal plants to manage diseases on the continent will be possible with the assistance of traditional health practitioners (THPs) in various African countries. Studies have shown that THPs on the African continent are always reluctant to provide information about the constituents of the herbal preparations they employ to manage diseases. The THPs think that scientific researchers make use of their knowledge without acknowledging their contributions to the development of indigenous medicinal plant research on the continent [46,100]. There is a need to ensure proper agreements that will ensure the protection of THP rights between all the stakeholders involved in the management of diseases on the continent. These intellectual property (IP) agreements between scientific researchers and THPs on the African continent must be respected. An exchange program (involving THPs and academia) to share ideas regarding the treatment of diseases among the various African countries should be encouraged.

### 3.4. Plant Parts Used in the Management of Diabetes on the African Continent

The most identified plant parts used in the management of diabetes on the African continent were as follows: leaves (45.55%), bark (14.44%), bulb (12.22%) and roots (8.88%) (Figure 3). This is incongruent with other studies around the globe that cite leaves as the preferred plant parts employed in the preparation of herbal medicines to manage diabetes [101,102]. The preferred application of leaves in herbal preparations could be because leaves pose a limited threat to the survival of the plant species in indigenous communities [103]. Hamel et al. [104] argued that the high patronage of the leaves in the preparation of herbal medicines is justified because of the chemical compounds stored in the organ as a result of the synthesis of secondary metabolites [104].

### 3.5. Methods of Preparation Used in the Management of Diabetes on the African Continent

The preferred methods of preparation are predominantly decoction (46.43%), infusion (25%) and maceration (11.90%) (Figure 4). This is in agreement with studies carried out in India in which decoction was the preferred method of the preparation of herbal medicines [105]. Similarly, Amal and Masarrat [106] reported that decoction was the preferred mode of the preparation of herbal medicines in a study conducted in Saudi Arabia. It is significant to point out that the preferred methods of the preparation of herbal medicines (mainly decoction and infusion) by traditional healers in the management of diabetes on the African continent could be due to its affordability and the lower amount of energy applied in terms of the preparation [107]. The limited use of roots in the preparation of herbal medicines, even though they are known to be rich in phytochemical constituents, is because their use poses a challenge to the survival of the indigenous medicinal plants in traditional African communities [108].

### 3.6. Route of Administration of the Medicinal Plants Used in the Management of Diabetes in Africa

Findings from this study revealed that the predominant route of administration is oral (52.32%) (Figure 5). This is like other studies that revealed that the preferred route of the administration of herbal medicines in the management of diabetes is the oral route [109,110]. Significantly, Lebbie et al. reported that the oral route of the administration of herbal medicines is a non-invasive and highly effective method [111].

## 4. Conclusions

This study demonstrates that there is an enormous diversity of indigenous medicinal plants utilized in the management of diabetes on the African continent. The highly cited indigenous medicinal plants in this study are *Azadirachta indica*, *Allium sativum*, *Moringa Oleifera* and *Olea europaea*. The medicinal plants documented in this study could assist researchers in developing novel medicines for the treatment of diabetes on the African continent and beyond. Fascinatingly, metformin, an oral hypoglycemic drug, was obtained from a medicinal plant (*Galega officinalis*) [112]. The contributions of the data presented in this study to the scientific community in our quest to find novel medicines to mitigate the effect of diabetes in our communities cannot be overstated. There is a need to conduct a comprehensive isolation of the biologically active components in the identified medicinal plants for pharmacological analysis purposes, and to determine their curative properties. To this end, extensive preclinical and clinical trials should be conducted using the medicinal plants frequently cited in this study to determine their antidiabetic potential. The employment of toxicological techniques to predict the safety of the isolated bioactive compounds from the documented indigenous medicinal plants in this study is highly recommended.

## Figures and Tables

**Figure 1 plants-13-01898-f001:**
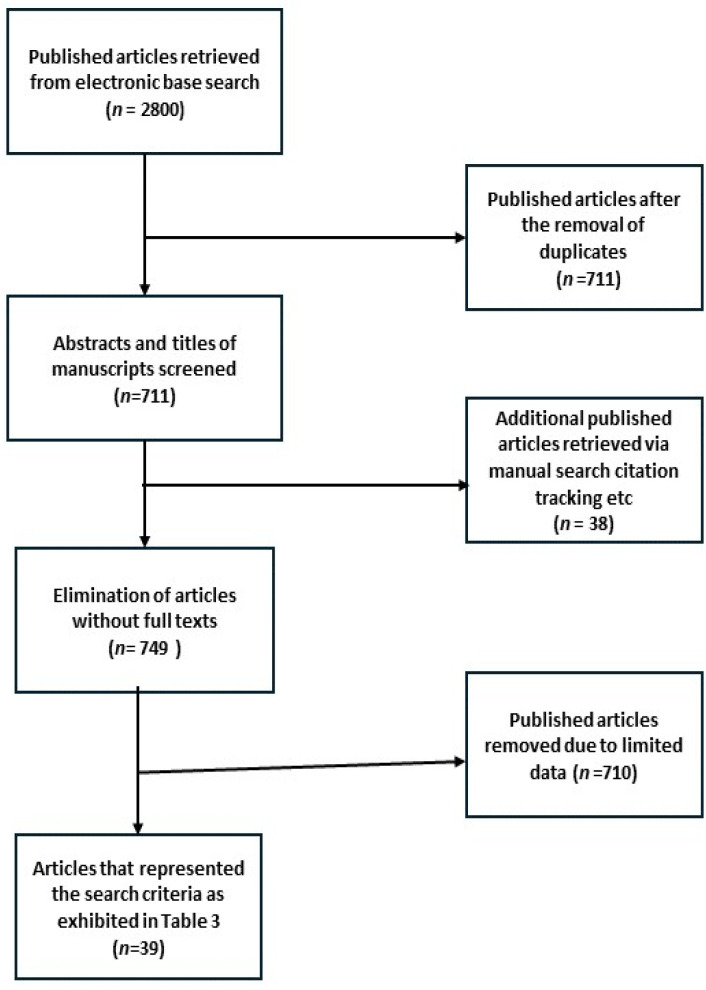
An overview of the procedure employed for the identification of the 39 published articles employed in this review.

**Figure 2 plants-13-01898-f002:**
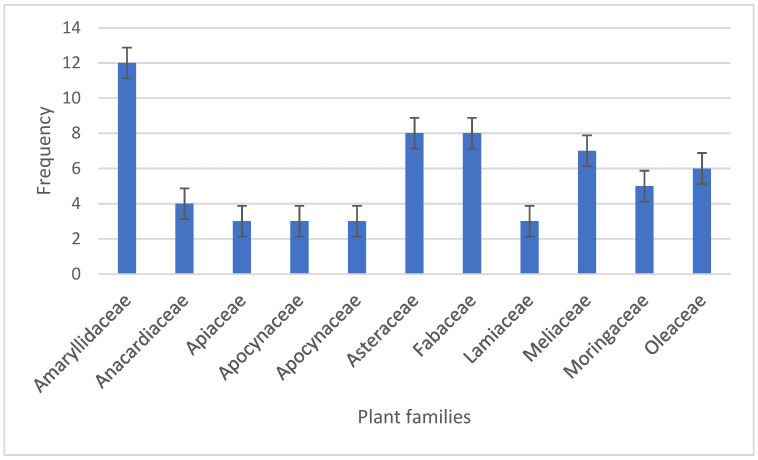
Frequencies (*n*) of plant families (*n* ≥ 3) utilized in the management of diabetes on the African continent. The rest of the plant families (*n* ≥ 2) are listed in Table 3.

**Figure 3 plants-13-01898-f003:**
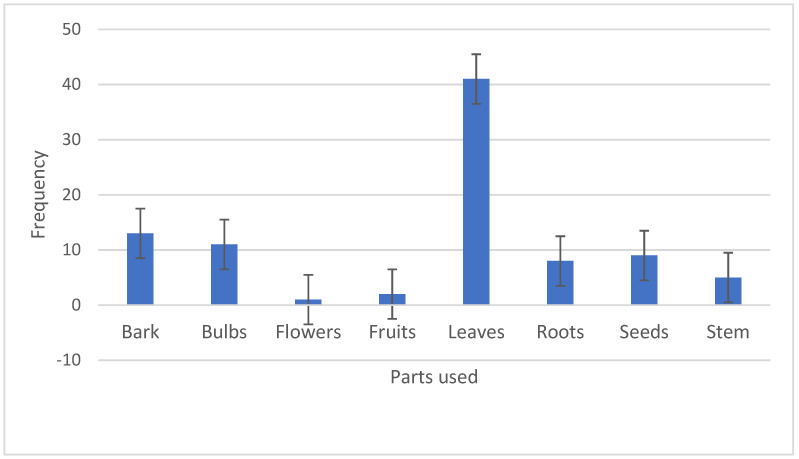
Frequencies (*n*) of the plant parts of the medicinal plants utilized in the management of diabetes on the African continent.

**Figure 4 plants-13-01898-f004:**
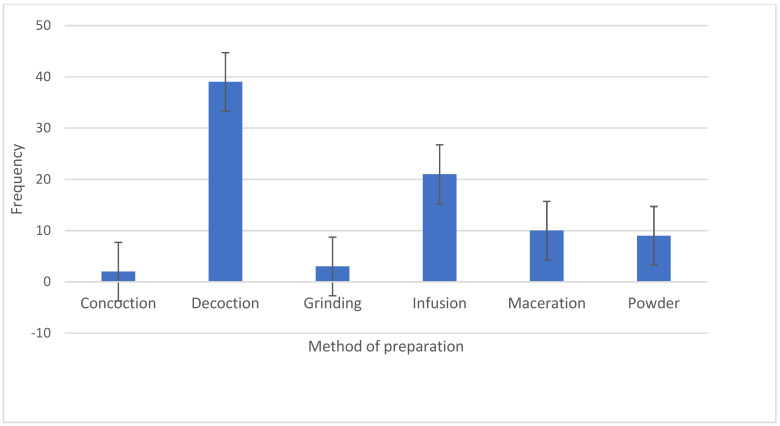
Frequencies (*n*) of the methods of preparation of the medicinal plants utilized in the management of diabetes on the African continent.

**Figure 5 plants-13-01898-f005:**
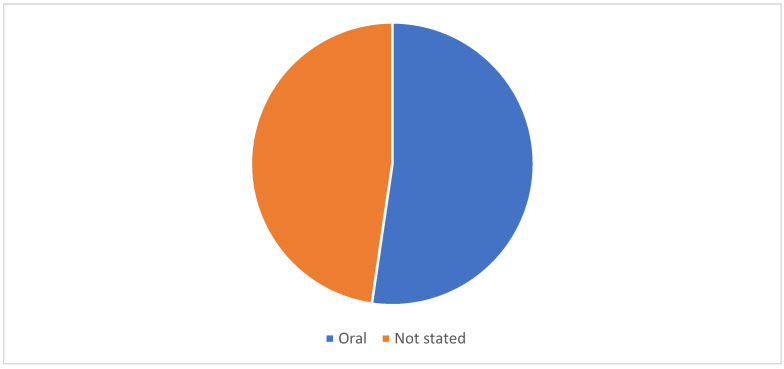
A pie chart showing the percentage of the route of the administration of the herbal medicines utilized in the management of diabetes on the African continent.

**Table 3 plants-13-01898-t003:** An overview of the indigenous medicinal plants utilized in the management of diabetes in Africa.

Botanical Name (Family)	Country (African Regional Bloc)	Local Name(s)	Part(s) Used	Method(s) of Preparation	Route of Administration	Reference
*Allium sativum* L. (Amaryllidaceae)	Nigeria (WAR)	Alubosa ayu (Yo)	Bulb	Grinding	NS	[44]
*Allium ascalonicum* L. (Amaryllidaceae)		Alubosa Elewe (Yo)	Bulb	Infusion	NS	
*Crinum jagus* (J. Thomps.) Dandy (Amaryllidaceae)		Ogede odo (Yo)	Bulb	Maceration	NS	
*Gymnanthemum amygdalinum* (Delile) Sch. (Asteraceae)		Ewuro (Yo)	Leaves	Maceration	NS	
*Azadirachta indica* A.Juss (Meliaceae)	Nigeria (WAR)	Dogon yaro (Ha)	Bark, leaves	Decoction	Oral	[45]
*Mangifera indica* Lam. (Anacardiaceae)		Mangoroo (Ha)	Bark, leaves	Decoction	Oral	
*Vernonia amygdalina* Delile (Asteraceae)		Shiwaka (Ha)	Leaves, stem	Grinding	Oral	
*Allium sativum* L. (Amaryllidaceae)	South Africa (SAR)	NS	NS	NS	NS	[46]
*Aloe barbadensis* miller (Asphodelaceae)		NS	NS	NS	NS	
*Trigonella* *foenum-graecum* L. (Fabaceae)	Algeria (NAR)	Halba (Ar)	Seeds	Infusion	Oral	[47]
*Artemisia herba-alba* Asso (Asteraceae)		Chih (Ar)	Leaves	Infusion	Oral	
*Allium sativum* L. (Amaryllidaceae)	Nigeria (WAR)	Tafarnuwa (Ha)	Bulb	NS	NS	[48]
*Azadirachta indica* A. Juss (Meliaceae)		Dogon yaro (Ha)	Leaves	NS	NS	
*Moringa oleifera* Lam. (Moringaceae)		Zogale (Ha)	Leaves	NS	NS	
*Allium cepa* L. (Amaryllidaceae)	Ghana (WAR)	Onion (Tw)	Bulb	Decoction	Oral	[49]
*Ampelocissus multistriata* (Baker) Planch. (Asteraceae)		Awonwono (Tw)	Leaves	Decoction	Oral	
*Morinda lucida* A. Gray (Rubiaceae)		Konkoroma (Tw)	Leaves	Decoction	Oral	
*Aloe barbadensis* miller (Asphodelaceae)	South Africa (SAR)	Mokgoba (Ts), Inhlaba (Zu)	Leaves	Decoction, concoction	Oral	[11]
*Hypoxis Hemerocallidea* Fisch., C.A. Mey & Ave-Lall (Hypoxidaceae)		Labatheka (Ts), inkomfe (Zu)	Roots	Decoction, concoction	Oral	
*Artemisia herba-alba* Asso (Asteraceae)	Morocco (NAR)	Chih (Ar)	Leaves	Decoction	NS	[50]
*Allium cepa* L. (Amaryllidaceae)		Elbessla (Ar)	Bulb, seeds	Infusion	NS	
*Allium sativum* L. (Amaryllidaceae)	Nigeria (WAR)	Ayuu (Yo)	Bulb	Decoction	Oral	[51]
*Vernonia amygdalina* Delile (Asteraceae)		Ewuro (Yo)	Leaves	Decoction	Oral	
*Coriandrum sativum* L. (Apiaceae)	Algeria (NAR)	Kosbor (Ar)	Seeds	Decoction, infusion	Oral	[52]
*Linum usitatissimum* L. (Linaceae)		Zeriêet el kettan (Ar)	Seeds	Infusion, powder	Oral	
*Olea europaea* L. (Oleaceae)		Zitoun (Ar)	Leaves	Infusion	Oral	
*Allium sativum* L. (Amaryllidaceae)	Zambia (SAR)	NS	NS	NS	NS	[53]
*Moringa Oleifera* Lam. (Moringaceae)		NS	NS	NS	NS	
*Moringa oleifera* Lam. (Moringaceae)	Nigeria (WAR)	Ewe igbale (Yo)	Leaves	NS	NS	[54]
*Azadirachta indica* A. Juss (Meliaceae)		Dongoyaro (Yo)	Bark, leaves	NS	NS	
*Allium cepa* L. (Amaryllidaceae)	Algeria (NAR)	Bassel (Ar)	Bulb	Grinding	Oral	[55]
*Trigonella foenum-graecum* L. (Fabaceae)		Helba (Ar)	Seeds	Infusion, decoction	Oral	
*Cinnamomum verum* J. Presl (Lauraceae)		Quarfa (NS)	Bark	Infusion, maceration, powder	Oral	
*Olea europaea* L. (Oleaceae)		Zeytoune (NS)	Leaves	Infusion, powder	Oral	
*Olea europeae* L. (Oleaceae)	Morocco (NAR)	Zaytoun (Ar)	Leaves	Infusion, powder	Oral	[56]
*Trigonella foenum-graecum* L. (Fabaceae)		Lhelba (Ar)	Seeds	Decoction	Oral	
*Thymus satureioides* Coss. et Bal. (Lamiaceae)		Z’itra (Ar)	Leaves	Decoction, infusion	Oral	
*Sida acuta* Burman f. (Malvaceae)	Tanzania (EAR)	Tevere (NS)	Leaves	Decoction	Oral	[57]
*Parinari* Planch.ex Benth. Curatellifolia (Rosaceae)		Mnazipori (NS)	Roots	Infusion	Oral	
*Opuntia ficus-indica* (L.) Mill., (Cataceae)	Zimbabwe (SAR)	Mudhorofiya (Sh)	Leaves	Decoction	Oral	[58]
*Cassia occidentalis* (Linn) (Fabaceae)	Sierra Leone (WAR)	E-sabe (Ti)	Leaves	Decoction	Oral	[59]
*Mangifera indica* Lam. (Anacardiaceae)		Mangro (Ka)	Leaves	Decoction	Oral	
*Moringa Oleifera* Lam. (Moringaceae)	Angola (SAR)	Moringa (NS)	Leaves, seeds, roots	Infusion, decoction	Oral	[60]
*Azadirachta indica* A. Juss. (Meliaceae)		Neem (NS)	Leaves	Infusion, maceration	Oral	
*Azadirachta indica* A. Juss. (Meliaceae)	Sudan (NAR)	Neem (Hd)	Roots	Maceration	Oral	[61]
*Kalanchoe glaucescens* Britten (Crassulaceae)		Malut (Hd)	Leaves	Infusion	Oral	
*Urtica dioica* L. (Urticaceae)	Morocco (NAR)	Herriga (thayzint) (Th)	Leaves	Decoction	NS	[62]
*Olea europaea* L. (Oleaceae)		Azamour (Th)	Leaves	Decoction, infusion	NS	
*Leptadenia hastata* (Pers.) Decne (Apocynaceae)	Ghana (WAR)	Yaadiya (Tw)	Leaves	Decoction	NS	[63]
*Pricralima nitida* Durand & Hook (Apocynaceae)		Abere (Tw)	Roots	Decoction	NS	
*Olea europaea* L. (Oleaceae)	Morocco (NAR)	Zitoun (Ha)	Leaves	Infusion	Oral	[64]
*Silvia officinalis* L (Lamiaceae)		Salmia (Ha)	Stem, flowers	Decoction, powder	Oral	
*Garcinia kola* Hackel. (Clusiaceae)	Democratic Republic of Congo (EAR)	Butenth (Wo), Nten (Mb)	Fruit, bark	Decoction, maceration	Oral	[65]
*Pancovia laurentii* (De Wild.) Gilg (Sapendaceae)		Irweth (Wo)	Bark	Decoction	Oral	
*Securidaca* *Longipedunculata* Fresen (Polygalaceae)	Nigeria (WAR)	NS	Stem, bark	NS	NS	[66]
*Pennisetum pendicellatum* Trin. (Poaceae)		NS	Stem, bark	NS	NS	
*Trigonella foenumgraecum* L. (Fabaceae)	Morocco (NAR)	Lhelbah (Ar)	Seeds	Decoction, powder	NS	[67]
*Verbena officinalis* L. (Verbenaceae)		Allouiza (Ar)	Leaves	Decoction, infusion	NS	
*Olea europaea* L. (Oleaceae)	Morocco (NAR)	NS	NS	NS	NS	[68]
*Crossopteryx febrifuga* (Afzel. ex G. Don) Benth. (Rubiaceae)	Tanzania (EAR)	Nakasabuni (Ya)	Roots, bark	Decoction	NS	[69]
*Maprounea africana* Müll. Arg (Euphorbiaceae)		Mtunu (Ya)	Roots	Decoction	Oral	
*Moringa oleifera* Lam. (Moringaceae)	South Africa (SAR)	NS	NS	NS	NS	[70]
*Abelmoschus esculentus* (L.) Moench (Malvaceae)		NS	NS	NS	NS	
*Ocimum basilicum* L. (Lamiaceae)	Mauritania (WAR)	I.hbaq (Hs)	Leaves	Decoction	Oral	[71]
*Ziziphus lotus* (L.) Lam. (Rhamnaceae)		Hreytek (Hs)	Leaves	Infusion	Oral	
*Opuntia ficus-indica* (L.) Mill., (Cataceae)	South Africa (SAR)	Toorofeye (Ts)	Leaves	Maceration	Oral	[72]
*Commiphora* *kerstingii* Engl. (Burseraceae)	Nigeria (WAR)	Ararrabi (Ha)	Leaves	NS	NS	[73]
*Khaye senegalensis* Desr. (Meliaceae)		Madachi (Ha	Bark	NS	NS	
*Hunteria umbellate* (K. Schum) Haller. F (Apocynaceae)	Nigeria (WAR)	Abeere (Yo)	NS	NS	NS	[74]
*Momordica charantia* Descourt. (Cucurbitaceae)		Ejinrin (Yo)	NS	NS	NS	
*Erythrina abyssinica* Lam. ex DC. (Fabaceae)	Uganda (EAR)	NS	Bark	NS	NS	[75]
*Ficus saussureana* DC. (Moraceae)		NS	Leaves	NS	NS	
*Allium sativum* L. Alliaceae	Morocco (NAR)	tūma, tiskert (Ar)	Bulb	Maceration, powder	NS	[76]
*Carum carvi* L. (Apiaceae)		Karwiyâ (Ar)	Fruit	Decoction, infusion	NS	
*Cassia sieberiana* DC. (Fabaceae)	Burkina Faso (WAR)	Sindjan (Di)	Roots, bark	Decoction, powder	Oral	[77]
*Chrysanthellum* *americanum* (L.) Vatke. (Asteraceae)		Waltuko (Mo)	Leaves	Decoction	Oral	
*Phaseolus Vulgaris* L. (Fabaceae)	Cameroun (CAR)	Evele kone (Bu)	NS	Maceration	Oral	[78]
*Aloe barbadensis* miller (Liliaceae)		Aloe vera (Bu)	Leaves	Decoction	Oral	
*Allium cepa* L. (Amaryllidaceae)	Uganda (EAR)	NS	Bu	NS	NS	[79]
*Mangifera indica* L. (Anacardiaceae)		NS	Leaves, bark	NS	NS	
*Allium cepa* L. (Amaryllidaceae)	Democratic Republic of Congo (EAR)	Litungulu (NS)	Bulb	Maceration	NS	[80]
*Mangifera indica* L. (Anacardiaceae)		Manga (NS)	Leaves	Decoction	NS	
*Persea americana* Mill. (Lauraceae)		Divoka (NS)	Leaves	Decoction	NS	
*Foeniculum vulgare* Mill. (Apiaceae)	Morocco (NAR)	Nafaa (Dr)	Seeds	Decoction	Oral	[81]
*Taraxacum officinale* L. (Asteraceae)		Garnina (Dr)	Leaves, roots, stem	Decoction, powder	Oral	

Local names of indigenous medicinal plants (specific to assigned countries) utilized in the management of diabetes in the various African countries: Arabic (Ar); Bulu (Bu); Dioula (Di); Darija (Dr); Hausa (Ha); Hadendowa (Hd); Hassani (Hi); Hassanya (Hs); Mbuu (Mb); Moore (Mo); not stated (NS); Shona (Sh); Tharifit (Th); Tswana (Ts); Twi (Tw); Wongo (Wo); Yao (Ya); Yoruba (Yo); Zulu (ZU). Regional blocs of African countries based on their locations on the continent: Central African Region (CAR); Eastern African Region (EAR); Northern African Region (NAR); Southern African Region (SAR); Western African Region (WAR).

## Data Availability

Data are contained within the article.

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
