# Peer review of "Indigenous Medicinal Plants Used in the Management of Diabetes in Africa: 5 Years (2019–2024) in Perspective"

_plants, 2024, doi:10.3390/plants13141898_

Round 1

Reviewer 1 Report

Comments and Suggestions for Authors

The paper entitled "Indigenous Medicinal Plants Used in the Management of Diabetes in Africa: 5 years (2019-2024) in Perspective ", is an overview of an absolutely important topic such as the use of medicinal plants in the management of human diseases and is indeed scientifically sound. Although the literature review is well organized,  in my opinion there are some major comments.

1. The hole text needs formatting (spacing, spaces between words, page boundaries). 

2. I have a major concern about  Table 1 and 2. The side effects of medication used for treatment of diabetes in not relevant to this study. It is misleading and should be excluded. It promotes a comparison between the Orthodox conventional medicine and medicinal plants used in management of human diseases. 

3. The paper overall is very difficult to read because of the very big Table 3. The data could be grouped or formatted, since there are comments on the table and figures summarizing the findings. 

4. The section 3.6 Route of Administration of the Medicinal Plants Used in the Management of Diabetes in Africa is not of great significance since oral administration is almost equal to not stated. 

Overall, the paper is interesting but my suggestion is that it should be more compact, with more discussion about the possible mechanisms of action of medicinal plants. 

Reviewer 2 Report

Comments and Suggestions for Authors

Authors explained their work in very simple way. Some suggestions for the authors. 

1. Authors should add a schematic work flow of the whole review. 

2. The tables format are not regular in the whole article. 

3. The graphs are without the error bar, please add the error bars. 

Comments on the Quality of English Language

English language is fine, few spell checks and the grammar should be checked. 

Reviewer 3 Report

Comments and Suggestions for Authors

The authors reviewed the literature (from 2019 to 2024)  for the Indigenous medicinal plants used in the management of diabetes in Africa.

The paper is well written, although the language need to be improved, and the authors - using their selection criteria - are reporting a number of articles (up to 41). 

I would reconsider after revision.

Comments:

Line 41: After “Shaw et al. “ provide the citation. The same for lines 185, 199, 203 and 242.

Lines 42-48: Rewrite for clarity

Line 46, 48, 63, 65: Use a common expression for “beta” or “β”  or “β-“ cells.

Line 71. Table 1 is not referenced in text

Line 81. Table 2 is not referenced nor commented in text

Paragraph 2.2. Step 1 is missing . The author are encouraged to provide each step and also comment how it was used in Figure 1.

Also in line 109 they report 41 articles whereas in the legend of Figure 1 (line 120) 39. Which is correct?

Par 3.1 lines 147-156: Provide the abbreviations for WAR, NAR etc .

Line 163- Table 3. Each row should have the country and the cited literature. Additionally, provide the abbreviation for NS. Moreover , additional details for e.g. possible side effect or cases used would be useful.

Line 175. Use italics for plant families

Lines 220-224. A Figure of table for common plant families used in Africa counties (following the authors’ observation reported in lines 216 to 218) and a possible comparison with other continents might be interesting.

Comments on the Quality of English Language

The paper is well written, although the language need to be improved

Reviewer 4 Report

Comments and Suggestions for Authors

This study reviews the indigenous medicinal plants used in the treatment of diabetes in the African continent. The existing results show that there is great diversity among the plants used for this purpose. The Indigenous medicinal plants that have received significant attention in this study are Azadirachta indica, Allium sativum, Moringa Oleifera, and Olea europaea.

The authors of this study have provided a general list of potential phytochemical constituents that have anti-diabetic properties. It is vital to identify the components of the natural products derived from the aforementioned Indigenous medicinal plants that are accountable for this specific activity. If there is a lack of existing data of this nature in the literature, then it is necessary to conduct a study of this nature when such data becomes available.

The study indicated that certain plants exhibit an anti-diabetic impact through the inhibition of alpha-amylase and alpha-glucosidase enzymes, while others demonstrate a high level of antioxidant activity. Given the existence of multiple methods of anti-diabetic action, it is imperative to elucidate these processes and identify the ingredients that promote each specific mechanism.

In my opinion, this study has serious flaws and I believe that it does not deserve to be published in the journal Plants.

Round 2

Reviewer 1 Report

Comments and Suggestions for Authors

Thank you for answering all my comments. 

Reviewer 3 Report

Comments and Suggestions for Authors

The authors replied to the comments raised. The revised version is suitable  for publication.

Reviewer 4 Report

Comments and Suggestions for Authors

By looking at the submitted revised version of the manuscript, it can be concluded that there was no significant improvement in the quality of the paper. Therefore, I stand by the original decision that the paper does not deserve to be published in this journal.
